# Device for Dual Ultrasound and Dry Needling Trigger Points Treatment

**DOI:** 10.3390/s23020580

**Published:** 2023-01-04

**Authors:** Gerardo Portilla, Francisco Montero de Espinosa

**Affiliations:** ITEFI-CSIC Spanish High Research Council, Serrano 144, 28006 Madrid, Spain

**Keywords:** ultrasound, physiotherapy, rehabilitation, dry needling

## Abstract

Ultrasound is a well-known tool to produce thermal and non-thermal effects on cells and tissues. These effects require an appropriate application of ultrasound in terms of localization and acoustic energy delivered. This article describes a new device that combines ultrasound and dry needling treatments. The non-thermal effects of ultrasound should locally amplify the needle’s effects. The ultrasound transducer can mechanically rotate in 3D space to align itself in the direction of the needle. The transducer electronically focuses the acoustic pressure automatically on the needle tip and its surroundings. A computer, using graphical interface software, controls the angulation of the array and the focus position.

## 1. Introduction

Physiotherapy and rehabilitation treatments with ultrasound have been performed for more than six decades. To date there has been no demonstrated agreement on the relationship between the doses applied in treatments and the therapeutic response in musculoskeletal diseases [1,2,3,4]. One of the problems demonstrating this disagreement is the way in which the published studies refer to a wide range of diseases and with varied doses of ultrasound. Ultrasound exposure produces various biological effects on cells and tissues. These are usually grouped into thermal and non-thermal effects to differentiate them since thermal effects were most-often used in the first decades of the use of physiotherapeutic ultrasound, due to the known relationship between energy absorption and the heating of the medium [5,6]. The mechanical interactions between the pressure wave and the medium [7,8], such as the cavitation that is produced—not only in cells but also in vivo—by ultrasound cause changes at the cellular level that are the cause of the cellular alterations that make possible the repair of musculoskeletal lesions [9,10]. One of the most commonly treated conditions is the presence of myofascial trigger points (MTPs) as they are a source of acute and chronic pain and can be disabling [11,12,13,14]. D. G. Simons et al. define MTPs as “a hyperirritable point in a tight band of skeletal muscle that is painful on compression, stretch, overload, or tissue contraction, usually responding with referred pain felt at a distance from the point” [15]. Some studies referring to MTP ultrasonic physiotherapy treatments conclude that the acoustic parameters of the treatment have been misused. They conclude that a higher ultrasonic intensity than that used in most studies is needed to achieve beneficial effects [16]. Alexander LD et al. published the results of a systematic review on ultrasound treatments of shoulder soft tissues. In this work, the authors propose the use of an acoustic energy density higher than 400 J/cm2, reiterating the futility of low-energy ultrasound claimed by some studies [17,18]. Another central question that tends to remain in the mind of an expert in acoustic propagation is: has the programmed acoustic pressure of the commercial or laboratory devices used always reached the right place at the desired acoustic pressure? The medium is a finite structure full of geometrically complicated reflectors, not a semi-infinite homogeneous material as assumed in the standards [19].

Dry needling (DN) is a treatment modality of chronic musculoskeletal pain used by physicians and physical therapists worldwide. It is based on the insertion of a thin needle through the skin to reach tender points in the body. Dry needling is defined by the American Physical Therapy Association as “a skilled intervention that uses a thin filiform needle to penetrate the skin, and stimulate underlying myofascial trigger points, muscular, and connective tissues for the management of neuromusculoskeletal pain and movement impairments” [20]. An excellent review of the history of DN was written by D. Legge [21]. Interestingly, the success of DN as a treatment of painful musculoskeletal disorders arose as result of an observation that the effects obtained with the injection of different substances on the muscles—anti-inflammation, anaesthesia—were similar to the use of needs on their own. In parallel with this fact, the explosion of the use of acupuncture to release pain in the 60’s ensured that DN treatments, guided by musculoskeletal scientific and contrasted theory, grew rapidly to be, at present, a well-known and contrasted therapy. The most targeted DN treatments are myofascial trigger points (MTP). DN of MTP is not only a local treatment from the point of view of repairing a stressed muscle structure but is also a treatment of a pain that is also reflected in points away from the MTP [22,23]. DN treatments have been shown to modify the composition of the chemical environment of the lesion and decrease the noise level of the endplates. However, the reason that an acupuncture needle produces this effect is not fully understood [24,25].

Deep DN treatments for MPTs produce different effects. Contraction knots can be broken, and contracted muscle bundles can be lengthened by reducing the overlap between actin and myosin filaments. Likewise, motor endplates can be destroyed by denervation the distal axons [25]. Mechanical, chemical, endocrinological, microvascular, neural and central effects have been reported [26].

It is interesting to note that in DN treatments the exact point of the MTP is localized when local twitch responses (LTR), which are involuntary spinal cord reflexes of the muscle fibres in a taut band, appear. This ensures that the tip of the needle is in the core of the MTP [27,28].

Few studies exist comparing DN with ultrasound treatments [29,30]. Interestingly, those studies that do exist compare DN with ultrasound physiotherapy that use ultrasound commercial systems but with high intensity levels. This ultrasound technique is call high-power pain threshold ultrasound (HPP-TUS) [31,32]. The treatment consists of the use of a continuous wave, and a gradual increase in intensity from the lowest intensity level to the intensity level at which the patient reports that pain is no longer tolerable, and then maintaining the sonication a few seconds more.

The clinical treatment that we propose consists of a two-step actuation. First, the physiotherapist introduces a needle to locate and treat the trigger point, performing well-known DN clinic procedures. Leaving the needle in the body, an ultrasonic device is then placed on the patient’s skin and delivers focused ultrasound at the tip of the needle and its surroundings to apply the programmed ultrasound treatment.

In the paper the design, realization and test of an ultrasonic device prototype to combine DN with ultrasound physiotherapy is described. The generation of non-thermal ultrasound effects at cellular level on the MPT and surrounding in the first stages of DN invasive treatment must amplify microvascular and tissue regeneration.

## 2. Materials and Methods

### 2.1. Piezoelectric Array Transducers

In a previous paper, the authors presented a new device to perform focused physiotherapy with two ultrasonic array transducer designs [33]. The same concept is used in this work, with the only difference being a hole that is made at the centre to permit the application of the array on the patient’s skin when the needle has been inserted in the body. Figure 1 shows the design of the two arrays showing the central hole. The array is made of concentric active elements that permit the electronic focusing of treatment along the propagation z-axis. The array, made of a 2D distribution of active elements, permits the electronic focusing of treatment across the entire 3D volume.

Two different array transducers were fabricated, one with eight concentric array elements and a 2D 28-element type. These were made with the aperture designs of Figure 1. The fabrication technique is described in [33]. Before fabricating the piezocomposite 1–3 discs, a bore was made at the centre of the piezoceramic discs with a diamond hollow drill—Diamond Board—using a high-speed drill—TV-4 Maquimetal.

### 2.2. Electrical Excitation

A commercial multi pulser with programmable excitation frequency, output voltage and channel excitation delays was used (SITAU, DASEL SL, Arganda del Rey Madrtid, Spain). The channel output electrical signal is a 0–150 V programmable unipolar negative square pulse. The pulser power source and the power transistors limit the number of cycles and pulse repetition rate. All the engineering and user software applications were made with LabView (National Instruments Inc., Austin, TX, USA).

### 2.3. Acoustic Field Simulation

COMSOL 5.3 (COMSOL Inc., Stockholm, Sweden) was used to simulate the vibration performance and the acoustic field of the transducers to take into account the non-homogeneous amplitude and phase vibration of the array elements and the appearance of structural resonance modes. The material parameters used in the simulation are the same as those used in Portilla et al. [33].

### 2.4. Transducer Test

The array transducers were tested. These tests including measurements of the input electrical impedance of the array elements, the mechanical vibration profile at the emission face in air and the acoustic diffraction field in water. The transducer elements input electrical impedance was measured with an impedance analyser (4294A Agilent, Santa Clara, CA, USA). The mechanical vibration pattern was tested in air coupling with a vibrometer (Polytec OFV 5000/505, Waldbronn, Germany). A water tank with computerized 3D cartesian stages—with a precision of 50 μm—and a PVDF needle hydrophone (Dapco V-19-T, Dexter, MI 48130, USA, 0,6 mm active head diameter) were used to perform the acoustic field measurements.

## 3. Results

### 3.1. Acoustic Field Simulation and Measurement

#### 3.1.1. Eight-Element Annular Array Transducer

After testing the vibration pattern of the arrays, the acoustic field of the two arrays was calculated and measured at different focus positions. Figure 2 shows, as example, the calculated (Figure 2a,b) and measured (Figure 2c,d) diffraction field in the zy plane of the eight- element array vibrating at f = 640 kHz at the programmed focus positions F(x,y,z)= (0,0,30) mm (Figure 2a,b) and F(x,y,z)= (0,0,50) mm (Figure 2c,d).

Figure 3 shows the comparison of the eight-element array calculated acoustic diffraction parameters with the experimental parameters. The calculations were made for six focus distances from F = 30 mm to F = 80 mm. The measurements include three focal distances F = 30 mm, F= 50 mm and F = 70 mm. Both the calculated and the measured acoustic pressure values are relative to the respective acoustic pressure maximum value. The measured relative acoustic pressure at the focus agrees better with the FEM simulation.

The focus position with respect to the programmed position and the –6 dB beam width as a function of the focus distance are well predicted. Focus length is larger than predicted. Imperfections in the construction of the transducer—such as adhesive interface with radius dependence thickness—and the consideration of the piezocomposite as a homogeneous material could be the reason for this disagreement.

Concerning the calculated absolute acoustic pressure and considering tissue attenuation a = 11.55 Np/m, pressure amplitude around 2 MPa can be ideally obtained at a depth range between 30 mm and 50 mm in muscle tissue with an electric excitation voltage V = 40 Vp.

#### 3.1.2. 2D 28-Element Array Transducer

The acoustic pressure diffraction field of the 2D 28-element array transducer was also calculated with the same attenuation coefficient a = 11.55 Np/m and electric amplitude voltage signal V = 40 Vp. Resonance frequency f = 860 kHz. The second row of Figure 4 shows the acoustic pressure diffraction at the focus positions F = 30 mm and F = 40 mm without electronic steering. Compared with the same programmed position focus for the eight-element array, the acoustic pressure level is more than four-time less. The reason, beyond the difference in diffraction aperture between concentric annular elements versus circular elements, is the decrease of active emission surface. Nevertheless, the 2D aperture design permits electronic steering. The third and fourth rows of Figure 4 show the electronically steered acoustic diffraction up to 10 mm along the y axis. As observed, pressure maximum decreases with the deflection angle—up to a 30%—and the focus distance—10%.

The ellipsoid length increases 80% with the focus distance as expected. The width also increases up to 50%. The increase of the deflection angle brings out the strength of the grating lobes at a volume closer to the transducer emission surface. This effect is larger for shorter focus distances.

The acoustic pressure diffraction was calculated when the 2D 28-element array transducer was steered electronically at two (x,y,z) focus positions: (5,5,40 mm) and (10,10,40 mm). The time delay for each array element was calculated for each focus and then converted on excitation voltage phase. The excitation voltage modulus was V = 40 Vp and an attenuation coefficient a = 11.55 Np/m was used. As seen in Figure 5a,c,d, referring to the non-steered case (Figure 5a), the pressure maximum at the focus locus decreases 15% at (5,5,40 mm)—7 mm from xyz = (0,0,40 mm)—and 35% at (10,10,40 mm)—14 mm from (0,0,40 mm). Grating lobes increase their relative amplitude from less than 28% without electronic steering up to 70% when steering at (10,10,40 mm). At the intermediate position (5,5,40 mm), the maximum amplitudes of the grating lobes have a relative pressure of 48% with respect to the focused acoustic pressure. If a sonication security zone is defined where the acoustic pressure is less than 6 dbs below the maximum lobe acoustic pressure [32], the (10,10,40 mm) electronic steering should be prohibited. The solution is to combine the mechanical and the electronic steering. The (10,10,40 mm) focus position can be attained moving first the transducer mechanically to the (5,5,40 mm) position, then electronically steering the pressure lobe another incremental (5,5,40 mm) (Figure 5e). Figure 5b,d,f show the corresponding experimental acoustic diffraction pressures of the simulated focus coordinates.

### 3.2. Device and Software

A new device was developed to apply a focused ultrasound at the tip of a DN needle introduced in the body following a simple procedure (Figure 6a). The device consists of a manipulator (Figure 6b, ②) with a needle angle IMU gauge (Figure 6b, ①) plugged into a computer, a multi-pulser, and a graphic interface software. The software was made in a format preferably compatible with Windows but can be adapted to any other operating system (Figure 7).

The manipulator has an actuator with a spherical parallel robot with two degrees of freedom to mechanically steer the array transducer (Figure 6d). This robot has two coplanar servomotors, SG90 Tower Pro, acting in a parallel mechanism, which rotates the ultrasonic array transducer in such a way that the acoustic diffraction lobe can be scanned in 3D space (Figure 6d, ⑦). A cavity filled with a gel in between the transducer array and the skin permits the coupling of the transducer as it is rotated (Figure 6d, ⑧).

The purpose of the actuator 3D rotation is to orientate the array transducer so as to locate it in the plane perpendicular to the needle axis. Therefore, a sensor is required to determine the angulation of the needle with reference to the manipulator application plane (XY plane of Figure 6). This sensor has a tube and an inertial sensor (Figure 6c, ③) connected to a microprocessor (Figure 6c, ⑤). The microprocessor is connected to a PC computer via USB. When the sensor tube is introduced in the needle section that remains out of the body after the puncture, the acceleration variables measured by the inertial sensor (*a_x_*, *a_y_*, *a_z_*) are calculated in the microprocessor and read by the computer, calculating the real inclination of the needle using Equation (1) [34,35,36]. To determine the position (x,y,z) of the needle tip, the microprocessor executes the rotation matrix Equations (2) and (3) with $z_{n}$ being the length of the needle section inserted into the body, a parameter that must be introduced previously in the software interface by the physiotherapist [37,38,39]. An action switch placed in the manipulator (Figure 6c, ④) activates the 2D rotation movement of the parallel robot orientating the array in the needle direction.
(1)$$\theta_{x}=arctan\left(\frac{a_{x}}{\sqrt{a_{y}^{2}+a_{z}^{2}}}\right)\;;\;\phi_{y}=arctan\left(\frac{a_{y}}{\sqrt{a_{x}^{2}+a_{z}^{2}}}\right)$$
(2)$$Rotx=\left(\begin{array}{ccc} 1 & 0 & 0\\ 0 & cos\;\theta_{x} & -sin\theta_{x}\\ 0 & sin\theta_{x} & cos\theta_{x} \end{array}\right);\;Roty=\left(\begin{array}{ccc} cos\emptyset & sin\emptyset & 0\\ 0 & 1 & 0\\ sin\emptyset & 0 & cos\emptyset \end{array}\right)$$
(3)$$\left[\begin{array}{c} x\\ y\\ z \end{array}\right]=Rotx\left(\theta \right).Roty\left(\phi \right)\left[\begin{array}{c} 0\\ 0\\ z_{n} \end{array}\right]$$

Once the coordinates of the needle tip are known, the software calculates the time delays of the electric excitation of all the array transducer elements to focus the ultrasonic field at the needle tip coordinates. The transducer frequency, propagation velocity and signal excitation parameters (excitation voltage, number of cycles, pulse repetition rate and treatment time) must be priorly introduced in the graphic interface software (Figure 7).

The graphic interface software has three XY charts with a red spot (Figure 7). The upper left chart is used to select the electronic (x,y) steering coordinates. A circle shows the limit to the appearance of acoustic pressure zones with a pressure larger than the 6 dB below the maximum of the acoustic pressure at the focus (Figure 7, ①). Two linear cursors can be moved to position the red point that represents the desired electronically steered focus position (Figure 7, ②). The bottom left chart is used to show the position of the acoustic focus when being steered mechanically.

The mechanical steering can be introduced manually for engineering test (Figure 7, ③) or automatically using the needle angle sensor for clinical purposes (Figure 7, ④). The upper right chart shows the result of the mechanical and electronic steering combination (Figure 7, ⑥). The value of the needle length inserted should be introduced manually (Figure 7, ⑦). The three green action buttons at the left activate pre-programmed ultrasonic treatments (Figure 7, ⑧). The clinic treatment can be started and stopped with a switch located in the manipulator or with software graphical switches (Figure 7, ⑨).

### 3.3. Clinical Procedure

Figure 8 shows the proposed clinical procedure. It consists of, firstly, the localization of the MTP in the patient’s body by the clinician (Figure 8a), MPT of 5 mm × 20 mm at 50 mm from the skin. Then, the needle is punctured through the skin and manipulated following known clinical procedures involving the performance of DN aiming to deactivate the MPT (Figure 8b). After the DN treatment, the needle must be left in the patient’s body such that its tip be as close as possible to the MPT’s volumetric centre. A thick layer of ultrasound gel should be applied to the skin around the needle (Figure 8c, ⑤). The next step is to place the transducer on the patient’s body, introducing the needle through the central transducer hole (Figure 8c, ④). The clinician must introduce the needle incision depth on the corresponding graphic interface window (Figure 7, ⑦). Next, the angle gauge is coupled to the needle and by pressing the “on” button of the manipulator, the transducer rotates mechanically (Figure 8d) and starts the programmed ultrasonic treatment, focusing the acoustic pressure on the needle tip (Figure 8e, ⑥). The acoustic diffraction lobe dimensions shown in the figure have those of the eight-concentric elements array focused at (0,0,50 mm). The manipulator should be maintained and fixed to the patient’s body during all the ultrasonic treatment.

As seen in Figure 8e the intersection of the acoustic pressure lobe has a pressure value higher than 6 dB below the pressure maximum, and the subject MTP does not cover the entire volume of the MPT. To sonicate the entire MTP volume, if necessary, the 2D array transducer is the most appropriate design because the acoustic diffraction lobe can then be electronically steered. As simulated in Figure 4 and Figure 8f,g, the acoustic pressure lobe can be deflected electronically from the needle tip position as much as ±10 mm from the focus location—needle tip—without decreasing the acoustic pressure more than 25%.

### 3.4. Mechanical Steering Accuracy Test

The mechanical angular steering accuracy of the device was tested with an experiment that reproduces the clinical procedure. A cylindrical piece was placed on the transducer hole, coaxial with the transducer’s z axis, to house on it a 650 nm laser diode with a lens—focus diameter of 3 mm and divergence angle of 0.019 degrees—illuminating in the forward transducer direction—(0,0,−z). A thin metallic bar coaxial with the laser beam was also housed in the rear direction—(0,0,z) direction. A screen with a millimetre scale was placed parallel to the manipulator emission plane at a distance of z = 500 mm. Different (x,y,500 mm) mechanically steered transducer positions are introduced in the corresponding software window (Figure 7, ③). By inserting the IMU in the metallic bar, the roll and pitch degrees are obtained. The microprocessor calculates the laser spot position in the (X,Y,500 mm) plane and the robot is then automatically rotated, moving the laser spot. The (x,y) positions of the laser spot in the screen are then measured. The absolute error of the measured position depends on the direction of the angulation—pitch and roll—indicating that the structure of the parallel robot has some construction imperfections. The error is larger for shorter angulations, which is compatible with the type of stepper motor used. The mean absolute error is Av(x,y) = (0.37,0.4 mm). Importantly, the error is less than 16% of the narrowest focus width tested with the 28-element, 860 kHz 2D array transducer.

## 4. Discussion

Needle angulation is a critical parameter. The larger the bore diameter, the smaller the emission aperture of the entire transducer. The smaller the bore diameter, the smaller the maximum needle angulation permitted. If the present transducer designs are taken as a reference, the maximum needle angle permitted is ±9 degrees for the case of the eight-element array—7 mm diameter centre hole and ±14 degrees for the case of the 28-eñement 2D array—10 mm diameter centre hole. Figure 8h,i show the angulation limit for the case of the eight-elements array prototype.

The acoustic diffraction of the concentric elements design presents a well-defined focus whereas the 2D design presents a focus zone and grating lobes that can sonicate the surroundings of the target with relatively high pressure. If no electronic steering is needed, because the focus covers the target, the concentric elements design is the best option. The calculated pressure maxima at the focus when exciting the array transducers with a monochromatic voltage V = 40 Vp are close to 2 Mp for the case of the eight-element design and close to 0.4 MPa for the case of the 28-element 2D array design. Voltage sources higher than 120 Vp should then be used if treatments aimed at applying short pulses of acoustic pressure higher than 1 MPa are needed [40,41].

Different array apertures can be designed to increase the acoustic pressure output efficiency. For instance, by using only a piezoelectric ring with a 40 mm external diameter and an aperture design of 48 sectorial elements of the same capsule design, the calculated acoustic pressure at the focus is more than twice that of the 28-element 2D prototype with the same excitation voltage.

As mentioned in [33], the objective pursued when developing focused ultrasonic systems for physiotherapy is to reach the area of the musculoskeletal lesion, and only this area, with the acoustic pressure necessary to promote thermal or non-thermal effects depending on the pressure and time-averaged intensity. Using a needle to reach the lesion, as is done when using DN, opens up the possibility of knowing the relative coordinates of the lesion in relation to the needle puncture location. By measuring the depth of the puncture and the polar angles of the needle, the lesion is then localized in 3D and the transducer can be oriented mechanically and/or electronically to deliver the desired acoustic treatment to the target. Moreover, the use of array transducers makes it possible to scan the lesion in a controlled manner. The combination of mechanical and electronic movement expands the available body volume for the safe delivery of the ultrasound dose while keeping the applicator fixed on the patient’s skin.

The ultrasound dose can be programmed during treatment, and the acoustic pressure, acoustic intensity and position can be guided with the user graphic interface. The proposed device is designed to obtain a relatively high-pressure ultrasonic to explore the action of non-thermal effects.

A parallel robot of the current prototype could be upgraded to achieve higher motion accuracy with better stepper motors and by machining the parallel mechanism instead of using 3D printing

## 5. Conclusions

A prototype device has been described to show how DN can be combined with focused ultrasound in a very simple way for use by physiotherapists and rehabilitators. All subsystems of this device mainframe—aperture and frequency of the transducer array, number of elements, acoustic pressure and delivered acoustic intensity, mechanical angulation of the transducer—can be redesigned according to clinical needs.

## Figures and Tables

**Figure 1 sensors-23-00580-f001:**
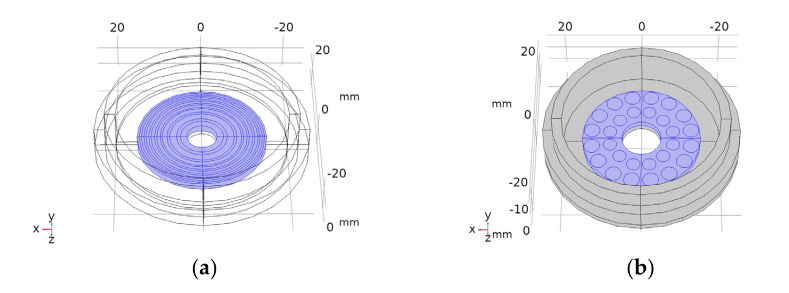
Drawings of the concentric elements array (**a**) and 2D 28-element array (**b**). Piezoelectric material in blue. Concentric circles (**a**) and circles in (**b**) are the piezoelectric active sections.

**Figure 2 sensors-23-00580-f002:**
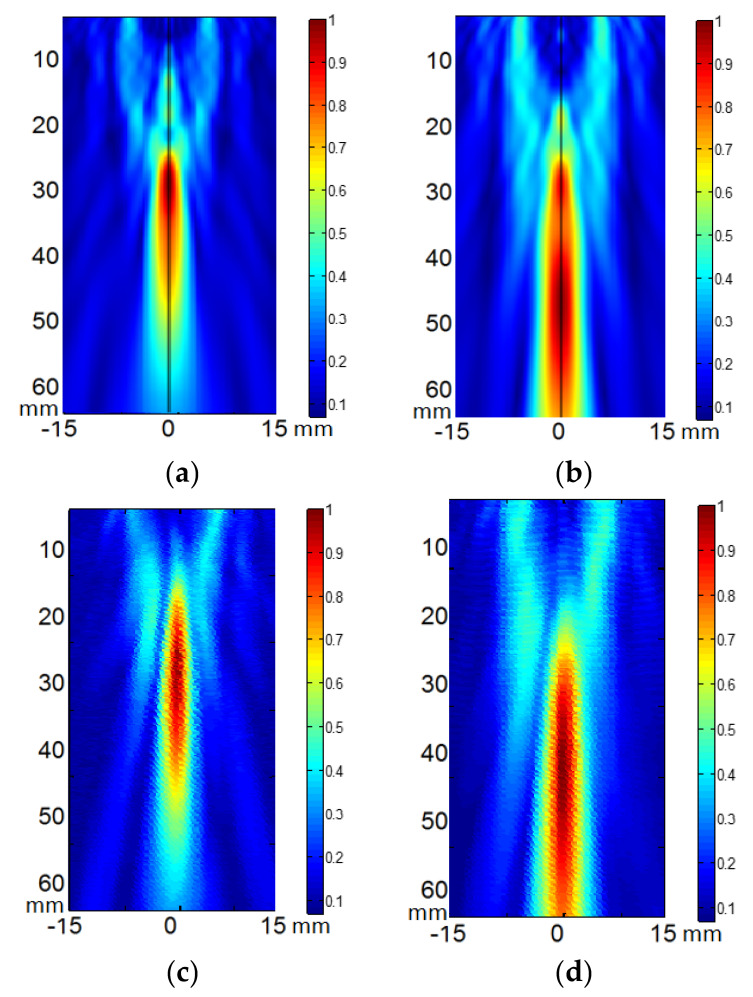
Acoustic pressure field in the zy plane of the eight-element annular array transducer when excited at 640 kHz. (**a**,**b**) calculated electronic focus at F(x,y,z) = (0,0,30) mm and F(x,y,z) = (0,0,50) mm respectively. (**c**,**d**) measured electronic focus at F(x,y,z) = (0,0,30) mm and F(x,y,z) = (0,0,50) mm respectively. Relative scales are used. Linear propagation and no attenuation were considered.

**Figure 3 sensors-23-00580-f003:**
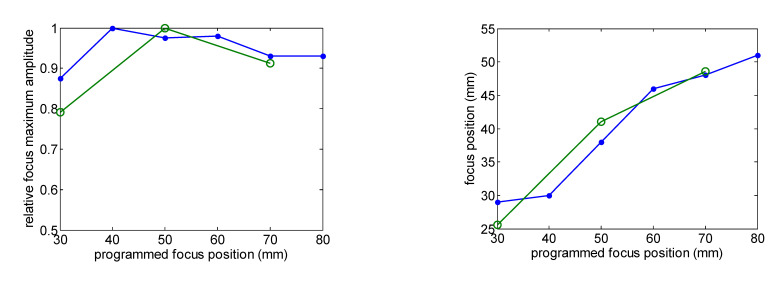
Comparison of the calculated (green line) and measured (blue line) acoustic diffraction parameters of the eight-element array. Frequency f = 640 kHz.

**Figure 4 sensors-23-00580-f004:**
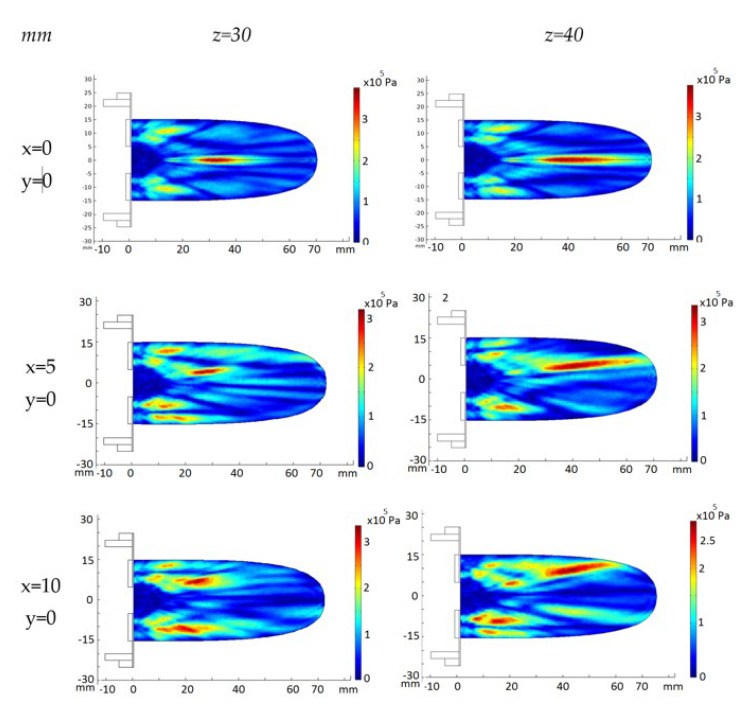
FEM_COMSOL calculated acoustic diffraction field of the 28-element 2D array. Resonance frequency f = 860 kHz. First column, (x,y) focus coordinates. Upper row, z focus coordinates.

**Figure 5 sensors-23-00580-f005:**
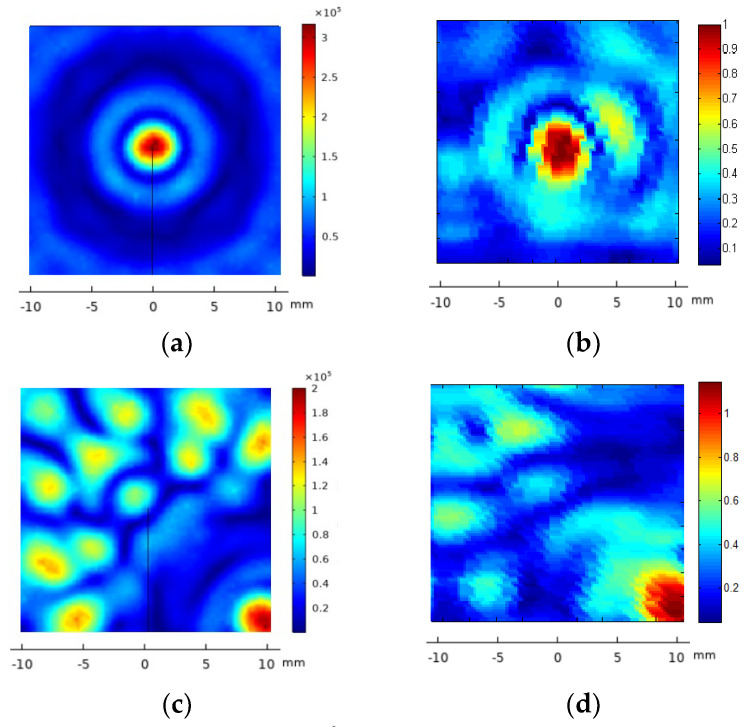
FEM_COMSOL acoustic pressure diffraction field of the 2D 28-element array excited at f = 860 kHz. XY plane. Left column, from top to bottom, the calculated acoustic diffraction in focus coordinates (x,y,z) = (0,0,40 mm)—(**a**), (x,y,z) = (10,10,40 mm)—(**c**) and a translation of coordinates (x,y,z) = (5,5,40 mm) from the focus case (x,y,z) = (5,5,40 mm)—(**e**). The right column, from top to bottom, shows the focused acoustic diffraction measured at (x,y,z) = (0,0,40 mm)—(**b**), (x,y,z) = (10,10,40 mm)—(**d**) and the result of combining the mechanical steering at (x,y,z) = (5,5,40 mm) and the electronic steering at (x,y,z) = (5,5,40 mm)—(**f**). The scale of the measured acoustic pressure is relative to the maximum pressure at (x,y,z) = (0,0,40 mm).

**Figure 6 sensors-23-00580-f006:**
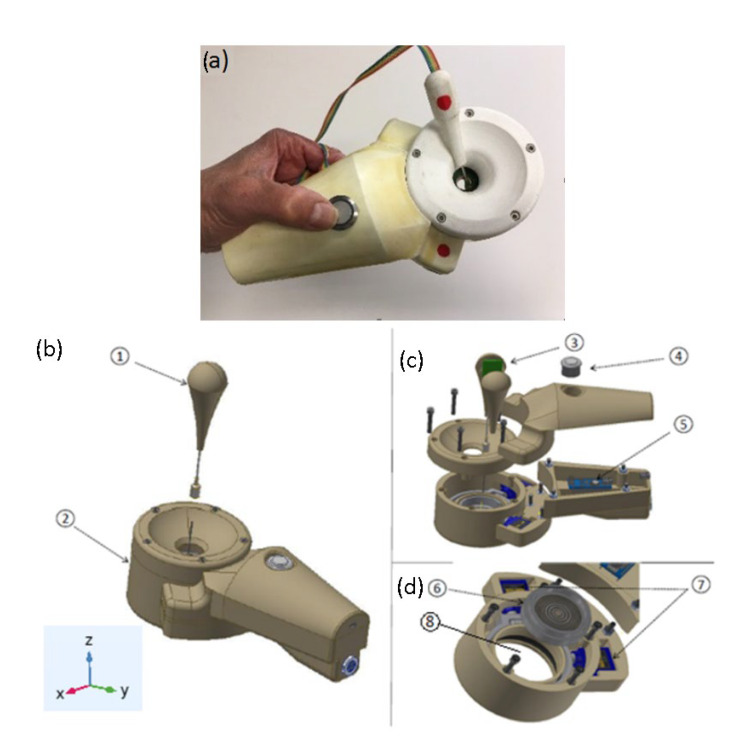
(**a**) Manipulator prototype with the needle angle gauge. (**b**) prototype description showing the needle angle sensor—①, the manipulator—②, and the reference geometric coordinates. (**c**) Inertial sensor IMU—③, main operation switch—④, and microprocessor—⑤. (**d**) Array transducer before being introduced in the parallel robot—⑥, parallel robot—⑦, and gel cavity—⑧.

**Figure 7 sensors-23-00580-f007:**
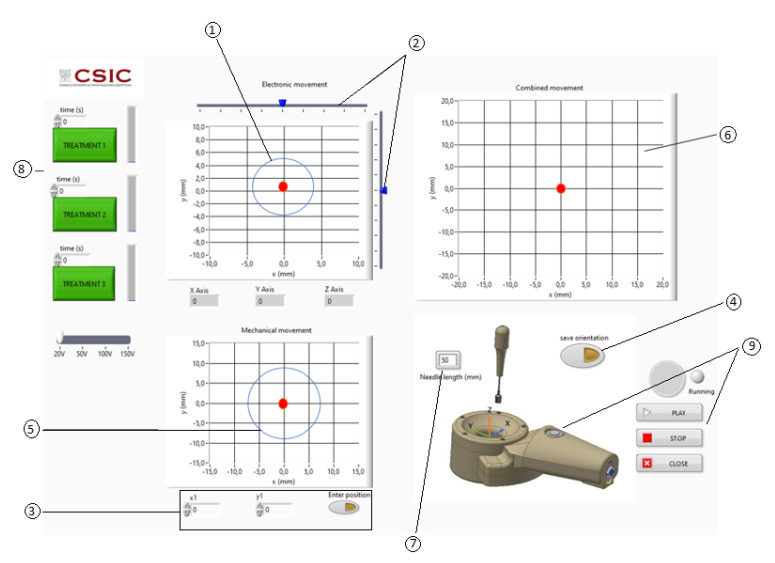
User graphic interface. ① Electronic steering: limit to obtain grating or diffraction lobes with an acoustic pressure larger than the 6 dB below the maximum of the acoustic pressure at the focus. ② Electronic steering linear cursors. ③ Manual introduction of the mechanical steering focus position. ④ Automatic positioning of the mechanical steering using the needle angle sensor. ⑤ Mechanical steering: limit to obtain grating or diffraction lobes with an acoustic pressure bigger than the 6 dB below the maximum of the acoustic pressure at the focus. ⑥ Combination of the mechanical and electronic steering. ⑦ Needle-inserted section. ⑧ Pre-programmed ultrasonic treatments. ⑨ Start and stop activation.

**Figure 8 sensors-23-00580-f008:**
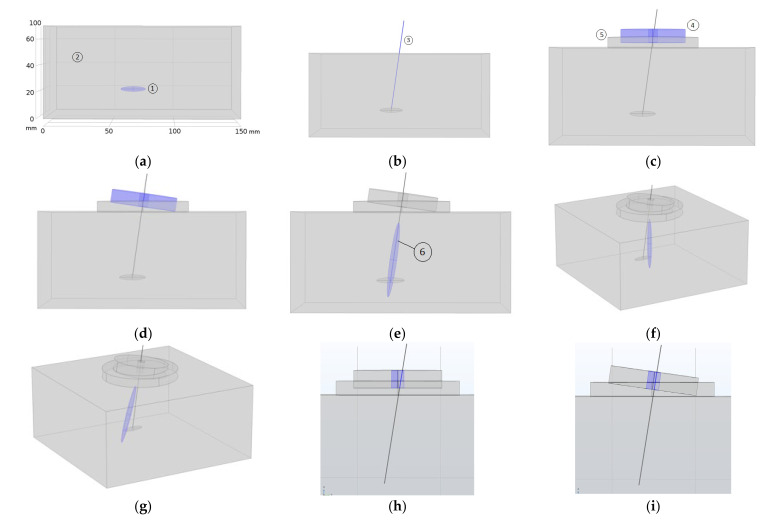
Graphical representation of the clinical procedure: (**a**) MTP localization—blue color-, (**b**) needle insertion, (**c**) transducer application—blue color-, (**d**) mechanical steering of the transducer—blue color- at the needle angular orientation, (**e**) focused sonication—blue color-, (**f**,**g**) acoustic lobe electronic steering—blue color-, and (**h**,**i**) maximum needle angulation permitted for the eight-element prototype.

## Data Availability

Not applicable.

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
