# Peer review of "Device for Dual Ultrasound and Dry Needling Trigger Points Treatment"

_sensors, 2023, doi:10.3390/s23020580_

Round 1

Reviewer 1 Report

This paper aims to develop a therapeutic ultrasound transducer combined with a dry needling technique. The authors designed a focused ultrasound transducer with the aid of numerical simulation, followed by the prototyping of the device. Their research is still in the developing stage, under clinical application, yet the research motivation and the potential research contribution seem promising. Though, the current form of the article still needs more help before the final publication. Therefore, in my opinion, this article can be published after some corrections. Please, refer to the specific review comments below.

1. For the Abstract, add one additional sentence to describe the potential advantage of the DN treatment combined with the ultrasound application.

2. For the title of the paper, what is the meaning of “Dual Ultrasound”?

3. For the last paragraph of the Introduction section, it would be great if you highlights the novelty and the significance of the research.

4. On page 2, “de needle”?

5. On page 3, it quotes “Figure 2 shows the real array transducer.” Is Figure 2 an acoustic pressure field rather than the schematic of the transducer?

6. On page 3, 3D cartesian stages- “50 m” precision- ?

7. A little confusing if the author claims the benefits of the 28-element transducer over the concentric one or just wants to compare the performances. In addition, the frequency of the concentric transducer is given by 640 kHz, whereas the frequency in the 28-element array is 860 kHz. Is this a fair comparison?

8. On page 4, it quotes, “eight focus distances from …” Is it eight, not six?

9. Add the alphabetic figure symbols (i.e., (a)-(d)) to the caption of Fig. 2 for a clearer description of each figure.

10. For Fig. 3, can you include any further discussion of the discrepancy between the test and the simulation results?

11. On page 5, what is “Vp”? peak-to-peak? Or zero to peak?

12. It is hard to identify the legend and the labels in Fig. 4. Modify the figure with a bigger size of fonts.

13. On the first line of page 6, “2D-24 elements…” Isn’t it 28?

14. Include how to steer the ultrasound beam in the simulation model.

15. Fig. 6(b) seems like redundant. How about including a schematic to explain the operation mechanism in Fig. 6(b) instead of the current figure?

16. On page 8, “senor”?

17. Where is Figure 10?

18. Include a more detailed explanation of Eq. (1) and (2).

19. In most captions of figures, the authors used “#)” to indicate the number shown in the figures. How about using the circled number in the caption to be the same as in the graphic representation?

20. For the conclusion section, the authors may separate the contents into 1) Discussion and 2) Conclusion.

21. Is there any technical limitations when the ultrasound is exerted around the solid needle? Any potential scattering of the acoustic waves?

Reviewer 2 Report

In this manuscriptthe authors developed a new device that combines ultrasound and Dry Needling treatments. The ultrasound transducer not only can rotate in 3D space mechanically to align itself in the direction of the needlebut also can electronically focus the acoustic pressure automatically on the needle tip and its surroundings. This work is so interesting that I recommend it to be published on Sensorsbut a minor revision is needed.

1、 As we all known, the frequency is an important parameter for ultrasound. Why do you select frequency below 1MHz? Please explain it.

2、 When 28 element 2D array was used, it is easy to get the ultrasound image of the needle fixed in the MTP. Why don’t you use ultrasound image to determine the position of the needle? If possible, please add some experiment about it.

3、 The purpose of such device development is therapy. What’s the therapeutic effect? Could you show us some related results?
